# The Impact of Visual Input and Support Area Manipulation on Postural Control in Subjects after Osteoporotic Vertebral Fracture

**DOI:** 10.3390/e23030375

**Published:** 2021-03-20

**Authors:** Michalina Błażkiewicz, Justyna Kędziorek, Anna Hadamus

**Affiliations:** 1Department of Physiotherapy, The Józef Piłsudski University of Physical Education in Warsaw, 00-968 Warsaw, Poland; justyna.kedziorek@awf.edu.pl; 2Department of Rehabilitation, Faculty of Medical Sciences, Medical University of Warsaw, 02-091 Warsaw, Poland; anna.hadamus@wum.edu.pl

**Keywords:** sample entropy, fractal dimension, center of pressure, postural control, osteoporosis

## Abstract

Osteoporosis is a prevalent health concern among older adults and is associated with an increased risk of falls that may result in fracture, injury, or even death. Identifying the risk factors for falls and assessing the complexity of postural control within this population is essential for developing effective regimes for fall prevention. The aim of this study was to assess postural control in individuals recovering from osteoporotic vertebral fractures while performing various stability tasks. Seventeen individuals with type II osteoporosis and 17 healthy subjects participated in this study. The study involved maintaining balance while standing barefoot on both feet for 20 s on an Advanced Mechanical Technology Inc. (AMTI) plate, with eyes open, eyes closed, and eyes closed in conjunction with a dual-task. Another three trials lasting 10 s each were undertaken during a single-leg stance under the same conditions. Fall risk was assessed using the Biodex Balance platform. Nonlinear measures were used to assess center of pressure (CoP) dynamics in all trials. Reducing the support area or elimination of the visual control led to increased sample entropy and fractal dimension. Results of the nonlinear measurements indicate that individuals recovering from osteoporotic vertebral fractures are characterized by decreased irregularity, mainly in the medio-lateral direction and reduced complexity.

## 1. Introduction

Osteoporosis (porous bone) is a disorder characterized by low bone mineral density (BMD) and structural deterioration of bone tissue. This condition leads to bone fragility and increased risk of fractures. Bone loss occurs silently and progressively, and so in many cases there are no symptoms until the first fracture occurs. Osteoporosis mainly contributes to fractures of the spine and proximal femur [1,2]. Two categories of osteoporosis have been described [3]—primary and secondary. Primary osteoporosis is the most common form of the disorder and includes postmenopausal osteoporosis (type I) and senile osteoporosis (type II). Type I is associated with a loss of estrogen and androgen resulting in increased bone turnover, with bone resorption. Type II presents as gradual age-related bone loss found in both sexes due to systemic senescence. It is induced by the loss of stem-cell precursors, with a predominant loss of cortical bone. Secondary osteoporosis is characterized as having a clearly defined etiologic mechanism (certain disorders or medical treatments). The World Health Organization [4] defines osteoporosis as the condition of having BMD or bone mineral content more than 2.5 standard deviations (SD) below the young adult mean normal value. Therefore, BMD correlates strongly with the risk of osteoporotic fractures.

Osteoporosis is a prevalent health concern among older adults and is additionally associated with muscle weakness and increased spine kyphosis leading to vertebral fractures [5,6]. A kyphotic posture may displace the center of gravity closer to the limits of stability, which means that the individual needs to make greater efforts to maintain balance even with small perturbations. Therefore, osteoporotic fractures lead to poorer balance and even falls. Weak muscles lead to impaired mobility, including reduced gait speed, greater difficulty climbing stairs [7,8,9], and choosing different strategies to maintain balance compared with what healthy people can easily manage [10]. Individuals with osteoporosis, with or without kyphosis, frequently prefer to use a hip strategy to maintain balance even when an ankle strategy would be more appropriate [10,11,12]. Additionally, it seems that postural stability in patients with osteoporosis has, until now, only been assessed using standard tools such as the center of pressure (CoP) path length, sway area, and mean sway velocity [6,13]. These tools unfortunately do not reveal changes occurring in the postural control mechanism. Therefore, in order to understand the physiology of postural control, nonlinear measures should be applied [14,15]. These measures make it possible to quantify the regularity, adaptability to environment, complexity, and efficiency or “automaticity”of postural control [14,15,16]. Nonlinear tools for evaluating the above-mentioned postural control properties include the largest Lyapunov exponent, Hurst exponent, recurrence quantification analysis (RQA), as well as fractal dimension and entropy families [14,15,17].

In this paper, the two most popular methods, sample entropy and Higuchi’s algorithm for computing fractal dimensions, were used to evaluate the CoP signal. The CoP signals are usually short time series, recorded mainly for 30s trials. It has been proven that both methods are well suited to the analysis of short time series [18,19,20].

Sample entropy is one of several types of entropy measures [19,21]. It shows great independence of data length [19]. Yentes, Hunt, Schmid, Kaipust, McGrath and Stergiou [18] proved that sample entropy starts to be sensitive for very smalldatasets (*n* ≤ 200). This measure is used to determine the regularity of postural sway and it quantifies the temporal structure of the signal. The calculation determines the probability of two similar sequences with the same number of data points remaining similar when another data point is added [22]. Increased values of sample entropy indicate a greater irregularity of the CoP value [15] and may be interpreted as an increase in the “automaticity”of postural control [14]. It has been proven that healthy people are characterized by higher values of sample entropy than persons with disabilitiesor injuries. Furthermore, the absence of vision led to a decrease in sample entropy as compared to when the eyes were open [21]. Complexity is crucial to flexibility in adapting to one’s surroundings. Thus, decreased complexity of physical movement translates into lower flexibility and higher rigidity in postural control [23]. Complexity can be measured in terms of fractal dimension [20]. To compute fractal dimension, many algorithms are available, such as Higuchi, Katz, or the box counting method [21]. However, only Higuchi’s algorithm can be applied to shorter time series without a loss of reliable values [20]. When applied to CoP displacement, a fractal dimension should lie between 1 and 2 [24]. Results which approach 2 indicate a high variability of sway. In the case of the CoP trajectory, a change in fractal dimension may indicate a change in control strategies for maintaining a quiet stance. It has been proven that loss of visual feedback while standing is accompanied by a significant increase in the fractal dimension of the CoP, which points to postural instability or the use of less stable control strategies [21]. Thus, the above-mentioned nonlinear dynamic measures, which quantify the structure or organization of the postural control variability along with linear measures describing the magnitude of the postural control variability, may be more sensitive for detecting postural control deficits and sensorimotor alterations associated with osteoporotic compression fractures of the spine. As such, the aim of this research was to assess postural control in individuals who have experienced osteoporotic compression fractures of the spine in comparison to healthy adults. Using nonlinear indicators as a tool should help to explain how the abilities of patients with osteoporosis to carry out individual balance tasks changes as the support surface area is reduced, visual control is switched off, and a dual-task is added. The following hypotheses were made:

**Hypothesis** **1.**
*Individuals recovering from osteoporotic vertebral fractures are characterized by decreased irregularity (lower values of sample entropy), reduced complexity, and less stable control strategies (lower values of fractal dimension) compared to healthy people;*


**Hypothesis** **2.**
*Individuals who have experienced osteoporotic compression fractures have a higher risk of falling.*


## 2. Materials and Methods

### 2.1. Participants and Procedure

Two groups took part in the research. The first group (OS) consisted of 17 persons with a mean age of 66.35 ± 6.13 years, body height 166.82 ± 11.11 cm, weight 71.82 ± 16.06 kg, and type II osteoporosis. All examined individuals had previously experienced osteoporotic compression fractures of the spine in the thoracic–lumbar region (low kyphosis). All of them had undergone minimally invasive surgery through the skin to insert perforated bolts reinforced with bone cement, which was performed 12 months before this study. All persons were operated on by the same surgeon. Exclusion criteria for the OS group included: unstable spinal fractures, impaired neurological status, vision impairments, inner ear disorder, use of an antidepressant, opioid or a sedative, current vertigo or dizziness, neurological disorder, peripheral neuropathy, and surgery in the last year that would cause balance deficits. It is worth mentioning that the patients had no noticeable foot dysfunctions or cardiovascular problems that could affect the test results.

The second group (C) consisted of seventeen healthy subjects with a mean age 42.88 ± 6.13 years, body height 167.19 ± 10.77 cm, and weight 70.86 ± 15.06 kg. The healthy people were a representative group, selected to ensure there were no statistically significant differences in the anthropometric parameters (e.g., body weight, height) in relation to the OS group. In addition, these individuals did not have any problems with balance. All participants gave their informed consent to take part in the study, which had previously been approved by the university’s institutional review board (no. 84/PB/2016).

Study participants underwent six balance measurements in the following order of three attempts lasting 20 seconds each, standing barefoot on both feet with eyes open (2eo), with eyes closed (2ec), and with eyes closed and dual-task (2ec_dt). A further three trials lasting 10 seconds each were taken during a single-leg stance balance task with eyes open (1eo), eyes closed (1ec), and eyes closed with dual-task (1ec_dt). The dual-task involved the participants being asked multiplication questions; taking into account performing calculations, only two- or three-digit sumswere asked during the trial, such as 11 × 15.

Data collection began after participants stated that they felt stable and ready to begin. A trial was discarded and then repeated if: (1) the non-tested limb made contact with the force platform or the stance limb; (2) the participant hopped or took a step with the stance limb; (3) the participant lifted a forefoot or heel. Center of pressure (CoP) trajectories in the anterior–posterior (AP) and medio-lateral (ML) directions were measured using the AMTI AccuSway force platform (Advanced Mechanical Technology Inc., Watertown, MA, USA) integrated with Balance Clinic software at a sampling rate of 100Hz. In addition, each subject underwent afall risk test (FRT) using the Biodex Balance System SD (BBS) (Biodex Medical Systems, Shirley, NY, USA). The BBS consists of a circular platform that is free to move in the AP and ML axes simultaneously. During the FRT test, the platform changed stability from very unstable to slightly unstable (from level 6 to 2). During the tests, the participants stood barefoot on the BBS support with arms held downward alongside the trunks of their bodies. For every one of the participants, this study was the first time they had ever encountered or been tested on an unstable platform.

### 2.2. Data Analysis

The study used the linear parameter of CoP path length and two nonlinear measures, sample entropy (SampEn) and fractal dimension (FD), to assess CoP dynamics. Path length values were exported from the AMTI plate and two nonlinear coefficients were counted, using MATLAB software (MathWorks, Natick, MA, USA), separately for medio-lateral (ML) and anterior–posterior (AP) CoP data, according to the rules described below. The data for the 10 s trials included 1000 points and for the 20 s trials, 2000 points in each direction.

#### 2.2.1. Stationarity of CoP data

All time series data were visually inspected for spikes and outliers. Experimental data were then examined for stationarity, using the MATLAB function for wide-sense stationarity estimation [25]. Approximately one-third of the trials demonstrated non-stationarity. In this paper, we used original, not fully stationary versions of the data, which was in line with the study [26,27].

#### 2.2.2. Sample Entropy (SampEn)

The Sample Entropy is mathematically computed as follows:
(1)From a vector $X=x_{1},x_{2},\ldots ,x_{N}$ two sequences of *m* consecutive points: $x_{m}(i)=x_{1},x_{2},\ldots ,x_{i+m-1}$ and $x_{m}(j)=x_{1},x_{2},\ldots ,x_{j+m-1},$
i,j∈[1,N−m],i≠j were selected to compute the maximum distance and compared to tolerance *r* for repeated sequence counting, according to: $$d[X_{m}(i),X_{m}(j)]=\max[\left|x_{i+k},x_{j+k}\right|]\leq r,\;(k\in [0,m-1],\;r\geq 0)$$
where the tolerance *r* is equal to 0.1∼0.2∗SD and SD is the standard deviation of *X_N_* [19].(2)$B^{m}(r)$ is the average amount of $B_{i}^{m}(r)$ for i∈[1,N−m] and $B^{m+1}(r)$ is the average of *m* + 1 consecutive points. Thus, sample entropy can be computed as follows:$$SampEn(N,m,r)=-\ln\left[\frac{B^{m+1}(r)}{B^{m}(r)}\right]$$

In the case of this paper, the SampEn was calculated using MATLAB codes obtained from the Physionet tool [28]. For calculating this measure, the “default” parameter values *m* = 2 and *r* = 0.2*(standard deviation of the data) were applied.

#### 2.2.3. Fractal Dimension (FD)

The FD was calculated using Higuchi’s algorithm [20]. Higuchi’s algorithm calculates the FD directly from time series. Reconstruction of the attractor phase space is not necessary; therefore, this algorithm is simpler and faster than other classical measures derived from chaos theory. Moreover, it can be applied to short time series [20]. Higuchi’s algorithm can be described as follows:
(1)For one dimensional time series: X=x[1],x[2],…,x[N], a new *k* time series can be formed as follows: $$X_{k}^{m}=x[m],x[m+k],x[m+2k],\ldots ,x\left[m+\mathrm{int}\left(\frac{N-m}{k}\right)\cdot k\right],$$
where *k* and *m* are integers, $\mathrm{int}\left(\frac{N-m}{k}\right)$ is the integral part of $\frac{N-m}{k}$, *k* indicates the discrete time interval between points, whereas *m* = 1, 2, …, *k*.(2)The length of each new time series can be defined as follows:$$L(m,k)=\left(\sum_{i=1}^{\mathrm{int}\left(\frac{N-m}{k}\right)}\left|x\left[m+ik\right]-x\left[m+(i-1)k\right]\right|\right)\frac{N-1}{\mathrm{int}\left(\frac{N-m}{k}\right)k^{2}},$$
where *N* is length of the original time series *X*.(3)The length of the curve for the time interval *k* is defined as the average of the *k* values *L*(*m*, *k*), for *m* = 1, 2, …, *k*:
$$L(k)=\frac{1}{k}\sum_{m=1}^{k}L(m,k).$$
Finally, when *L*(*k*) is plotted against 1/*k* on a double logarithmic scale, with *k* = 1, 2, …, *k_max_*, the data should fall on a straight line, with a slope equal to the FD of X. Thus, Higuchi’s FD is defined as the slope of the line that fits the pairs $\left(\ln[L(k)],\ln\left(\frac{1}{k}\right)\right)$ in a least-squares sense. In order to choose an appropriate value of the parameter *k_max_*, Higuchi’s FD values were plotted against a range of *k_max_*. The point at which the FD plateaus was considered a saturation point, and that *k_max_* value should be selected [29]. A value of *k_max_* = 100 was chosen for our study.

Statistical analyses were performed using PQStat v.1.8.2 (PQStat Software, Poznań, Poland), with the significant *p*-value set at 0.05. All coefficients were tested for normal distribution, using the Shapiro–Wilk test. The nonparametric Mann–Whitney U test with Z-statistics was used in order to compare whether there is a difference between the OS and C group for all nonlinear and linear parameters. Next, within the control group and OS group, the Skillings–Mack test as a nonparametric version of repeated measures ANOVA [30] and post hoc Dunn test with Bonferroni correction were applied in order to examine differences between the trials for all coefficients. In the OS group, not all subjects completed the remaining measurements; therefore, it is reasonable to use the above-mentioned test.

#### 2.2.4. ANOVA Skillings–Mack (Missing Data)

Skillings and Mack [31] proposed their “Friedman-Type” statistic as a general method for comparing treatments for an unbalanced/ incomplete block design or in the presence of missing data. Data can be missing by design or missing completely at random. However, each object must have at least two observations. Similar to the Friedman test, the responses are ranked within the block, but Skillings and Mack weigh the blocks to account for the unbalanced design. If there are no associated ranks and there are no deficiencies, it is the same as the Friedman ANOVA, although if missing data are present in a balanced system, it corresponds to the Durbin ANOVA results [30].

## 3. Results

### 3.1. Increasing Task Difficulty

Table 1 shows how increasing the difficulty of individual stability tasks affects how well healthy people and those with osteoporosis were able to perform them. The tasks were completed from the simplest (2eo) to the most difficult (1ec_dt). Healthy people completed all tasks, while people from the OS group were not able to complete all tasks, especially while standing on one leg. The results show that as the balance exercises become more difficult, fewer people from the OS group were able to complete them. In the study, those who did not pass the trial for the first time refused to repeat it, despite encouragement from the researcher.

### 3.2. Nonlinear Dynamics Indicators

#### 3.2.1. Analysis between Groups

Starting the analysis with a classical comparison of the CoP path length in individual attempts, we noted a tendency of the path lengthening proportionally with the difficulty of the task in both groups. Closing the eyes, with the absence of changing in the base of support, increased the CoP path length in both groups. However, a significant change was recorded only in the group of healthy persons while standing on one leg. The reduction in the base of support while maintaining the same visual conditions caused a significant increase in the CoP path length in both groups. It should be noted that the CoP path length values in the OS group were higher than in the C group, but a significant difference (U = 73, Z = 2054, *p* = 0.039) was noted only for the 2ec trial (Figure 1).

Differences in variance between groups were found in SampEn (*p* = 0.038) and FD (*p* = 0.031) in the frontal plane (x) for 2ec_dt task. Comparing group C with OS, significantly higher values of nonlinear parameters were noted in C group only for: SampEn and FD in frontal plane (x) for 2eo (SampEn: U = 59, Z = 2182, *p* = 0.028; FD: U = 32, Z = 3305, *p* < 0.001) and 2ec tasks (SampEn: U = 72, Z = 2092, *p* = 0.035; FD: U = 73, Z = 2054, *p* = 0.039) (Figure 2a,b).

#### 3.2.2. Analysis within Groups

In the OS group, a significantly (*p* = 0.045) longer path was recorded in the 1eo task than in the 2eo task (Figure 1), even though the duration of the 1eo test was half of that of 2eo. 

Comparing the nonlinear factors between the directions, it should be noted that the values of most of them were higher for the frontal plane (x). For the sagittal plane (y), non-significantly higher values were only obtained for SampEn_2ec_dt and SampEn_1eo. Following the Dunn post hoc test, statistically significant differences were observed between 2eo and 1eo for all parameters, except SampEn in direction x. These coefficients were significantly (*p* = 0.007, *p* = 0.010, *p* = 0.001) lower for 2eo, respectively, for SampEn_y, FD_x and FD_y. Additionally, a statistically significant difference was noted for FD_x and y between 2ec and 1eo (*p* = 0.023, *p* = 0.036, respectively). The values of FD were significantly lower for the 2ec trial. It should be stressed that the values of all nonlinear coefficients increased relative to the original task of standing on both feet with eyes open (Figure 2). In summary, in the OS group, elimination of visual control increased the value of all nonlinear parameters. The reduction in the base of support causing these increases was significant, except for SampEn_x (increase was not significant).

In C group, a significantly longer path was recorded during the 1eo, 1ec and 1ec_dt tasks than in the 2eo task (Figure 1). A significantly longer path was also observed in the 1ec and 1ec_dt tasks in relation to the 1eo, 2ec and 2ec_dt tasks. As in the OS group, it should be noted that the values of the majority of nonlinear dynamics factors were higher for the frontal plane (x) (Figure 2). Significant differences between directions were noted for SampEn and FD for the following tasks: 1eo, 2eo, 2ec. Following the Dunn post hoc test with Bonferroni correction for Skillings–Mack, statistically significant differences were observed between 2eo and 1eo for all parameters, except SampEn in the x direction. All nonlinear dynamic coefficients were significantly lower for 2eo. The same result was noted for the OS group. Additionally, significant lower values of SampEn_y, FD_x and FD_y were noted for the 2ec task compared to 1eo. Moreover, for the 1eo task, FD_x was significantly higher than that obtained during the 2ec_dt task. Further significantly higher values were noted for the parameters: SampEn_y and FD_y during 1ec and 1ec_dt tasks in comparison to those recorded for trials 2eo, 2ec. For the entropy calculated in the y direction, significantly lower values were additionally shown for the 2ec_dt task, compared to 1ec and 1ec_dt trials. In group C, all differences found for nonlinear and linear parameters were in the range of 0.001–0.03 significance levels. In summary, in the C group, elimination of visual control during both-leg standing non-significantly increased the value of SampEn and FD in the y direction. On the other hand, for these parameters in the x direction, there were noted insignificant decreases, although the elimination of visual control when standing on one leg caused no significant decreases in the values of SampEn_x, FD_x and FD_y. As in the OS group, the reduction in the base of support caused significantly increased values of all nonlinear parameters with the exception of SampEn_x (insignificant increase).

### 3.3. Fall Risk Test and Summary of Balance Test Results 

Table 2 shows the results of the fall risk test (FRT) for the OS group, where 10 subjects obtained very good scores. Four individuals were below the norm and six were in the norm for their age group. Seven subjects had poor results (i.e., fall), which meant that people in this group had a high risk of falling. In the group of healthy persons, ten were in the norm and seven were below the norm for their age group. It should be noted that a high risk of a fall on the BBS platform is not an indicator of the subject’s ability to complete stationary tests. Two individuals at a high risk of falling completed four static tasks, while three subjects who were not at high risk were able to complete just three static tasks, which involved standing on both legs. As such, it appears that measurements of postural stability under dynamic conditions, as taken using the BBS platform, will not be reflected in the results of tests under static conditions for the same cohort.

## 4. Discussion

The objective of this study was to assess postural control in individuals who had undergone osteoporotic spinal fractures. We used nonlinear measures to assess their ability to complete specific balance tasks while increasing the difficulty level by reducing the base of support, eliminating visual control, and adding a dual-task under static conditions. Postural stability was also assessed under dynamic conditions, using the Biodex Balance plate, in order to assess the risk of falling. We hypothesized that individuals who have experienced osteoporotic compression fractures have a higher risk of falling. In this study, ten individuals out of an OS group of 17 obtained good scores in the fall risk test, including four who were below the norm and six who were in the norm for their age group. Seven individuals obtained poor results, linked with a high risk of falling. Results on a tilting platform were not correlated with static test results. Two individuals at a high risk of falling completed four static tests, including all those involving standing on both legs and one standing on one leg with eyes open. Three other subjects completed a combination of tests while standing on both legs without any problems. Three individuals with very good FRT results (within the norm) were unable to complete the tests while standing on one leg. It is worth mentioning that in the control group, ten persons were in the norm and seven were below the norm for their age group. Decreased postural stability, measured in dynamic conditions in patients with osteoporotic vertebral compression fractures, was confirmed by Wang, Liaw, Huang, Lau, Leong, Pong and Chen [12]. Thus, we can stipulate that the FRT assesses balance as well as flexibility of the body. Balance assessment in static conditions may also be an inadequate tool to evaluate fall risk. Some studies reported that dual-task tests during the static test adds value to fall prediction in the elderly [32], although subjects having difficulty with multiple tasking may prioritize performance on the balance task and accept a decline in the cognitive task [33]. Such dependence was not evaluated in our study. On the other hand, fall risk factors include not only decreases in dynamic or static balance assessment, but also differences in gait, muscle strength, and fear of falling, according to their experience of falls [34]. Therefore, fall risk assessment performed in our study is indicative, although on the basis of the presented results, it can be concluded that the predisposition to falls in this group is moderate.

It should be stressed here that all individuals from the OS group had suffered spinal fractures in the past, which, according to Cunha-Henriques et al. [35], reduces the flexibility and mobility of the trunk and leads to reduced bodily complexity. Kinsner [36] defined complex systems as structures composed of many components which may interact with each other. They are systems whose behavior is intrinsically difficult to model due to the dependencies, competitions, relationships, or other types of interactions between their parts, or between a given system and its environment. Systems exhibit distinct complexity properties that arise from these relationships, such as nonlinearity, emergence, spontaneous order, adaptation, and feedback loops. One example of such a complex system is postural control—the mechanism by which the central nervous system regulates sensory information from other systems in order to produce adequate motor output to maintain a controlled, upright posture. The visual, vestibular, and somatosensory systems are the main sensory systems involved in postural control and balance [37]. Furthermore, when a sensory input is removed or disturbed, impairment of postural control increases.

In our study, as the balance tasks became more difficult, fewer participants from the OS group were able to complete them. When standing on both feet, the elimination of visual control resulted in one individual being unable to complete the task. When the dual-task was added, another individual was not able to complete the exercise. While the subjects were asked to stand on one leg while keeping their eyes open, another seven individuals were unable to complete the task. Meanwhile, when visual control was eliminated, a further six subjects were unable to complete the task. None of the subjects were able to stand on one leg with eyes closed and complete the dual-task. In the control group, such problems were not observed. It is worth noting that although the duration of the balance task involving standing on one leg was half that of the task involving standing on both legs, the CoP path length was significantly longer. Thus, in both groups we observed a tendency of the CoP path length to extend proportionally with the difficulty of the task, as previously noted by Abreu et al. [38]. However, adding a cognitive task did not change the values of the CoP path length significantly. Its values increased by 2.3% and 3.8%, respectively, in the OS and C group for both-legs standing and by 1.3% for one-leg standing in group C. It is worth emphasizing that in the studied groups, the elimination of visual control while maintaining the same base of support caused an insignificant increase in CoP path length, while the reduction in the base of support while maintaining the same visual conditions caused a significant increase in the CoP path length in both groups. Therefore, it seems that reduction in the base of support is a more difficult task in both groups. Nevertheless, da Costa et al. [39] point out that an increase in linear values does not always suggest a lack of balance, but rather a certain strategy on the part of skillful individuals to explore their support base by being more flexible. Additionally, the sensation and integration of sensory inputs and neuromuscular control deteriorates with increasing age and with the presence of osteoporosis, which increases neuromuscular noise [40]. Furthermore, the mechanical properties of passive structures such as tendons and ligaments also change with age and various disorders. According to the loss of complexity theory formulated by Lipsitz and Goldberger [41], aging is related to a loss of complexity in biological signals with a corresponding loss of adaptability. Stergiou et al. [42] formulated a theory of optimal movement variability, suggesting that the loss of complexity in movements can be characterized by both reduced regularity (i.e., towards a random pattern) and increased regularity (i.e., towards a periodic pattern). As such, in order to find out the type of postural control used by individuals with osteoporotic spinal fractures, non-linear variables of sample entropy and fractal dimension were used. We also hypothesized that individuals recovering from osteoporotic vertebral fractures are characterized by decreased irregularity (lower values of sample entropy) and reduced complexity and less stable control strategies (lower values of fractal dimension) compared to healthy people.

Biological data often present non-stationarity (e.g., a drift). Stationarity was not originally set as a requirement for SampEn [19]. The exact nature of stationarity required for SampEn, as well as for other non-linear measures, is unclear [26]. Nevertheless, several authors suggest that the time-series signal should be differenced prior to the calculation of non-linear measures [43,44,45]. It has, however, been shown that differencing had a strong impact on the data. Lubetzky et al. [27] and Rhea et al. [46] found that values of SampEn were about 3–4-fold higher than those for original data. Moreover, most of the differences observed with the raw data have changed. It is also worth emphasizing that Stergiou [26] suggested to remove trend or drift only if there is a strong reason to believe that drift is not part of the human performance pattern. Thus, in accordance with the above, non-linear measures were calculated for the raw data.

In the OS group, the lack of visual information resulted in increased sample entropy and fractal dimension in both anterior–posterior and medio-lateral directions while standing on two legs. However, these increases were not statistically significant. On the other hand, in the group of healthy people, elimination of visual control caused a decrease in the values of sample entropy and FD in the medio-lateral direction while standing both on one and two legs. Introducing the dual-task to the task with eyes closed resulted in increased values of all coefficients with respect to the baseline (2eo) in both groups, while no changes were seen with respect to the 2ec test. When the base of support was reduced (1eo), we observed a statistically significant increase in the values of all coefficients in both planes in relation to 2eo in both groups, except the sample entropy in the frontal plane, where the increase was not significant. The task involving standing on one leg with eyes closed (1ec) was completed by just one individual from the OS group, while the values of all nonlinear coefficients were the highest. In the control group, the task of standing on one leg with eyes closed as well as with dual-task was completed by all persons. It is worth emphasizing that adding a dual-task did not cause significant changes for nonlinear parameters in relation to standing with eyes closed, both when standing on one leg and on two legs. Comparing both groups, it can be noticed that significant higher values of nonlinear parameters were noted in the control group only for SampEn and FD in the frontal plane (x) for the 2eo and 2ec tasks.

It should be stressed that sample entropy quantifies signal regularity and complexity [47]. High entropy may indicate increased complexity and hence signs of a healthy vigilant system, or it may be interpreted as an ineffective attentive control of balance [48]. Lower sample entropy values show that the CoP signal is more regular and predictable, which is associated with less complexity of structure [49]. Complexity is crucial to flexibility in adaptation to the surroundings; therefore, this lower complexity of physical movement translates into lower flexibility and higher rigidity of postural control [23]. It is worth emphasizing that in the group of individuals with osteoporosis, the values of sample entropy were lower only in the frontal plane in all tasks and significantly lower only during both-leg standing with eyes open and closed in comparison to the control group. The same tendency was noted for fractal dimensions. In the sagittal plane, the values of nonlinear parameters were non-significantly higher in the group of persons with osteoporosis in comparison to the control group. Benjuya et al. [50] and Hsu et al. [51] suggested that lower SampEn observed in older adults could have been due to the utilization of greater muscle co-activation or joint rigidity as their postural strategy. Therefore, in cases of individuals recovering from osteoporotic vertebral fractures, lower SampEn represents a more regular pattern for CoP variability, which is the rigid strategy, and lower adjustment to perturbations mainly in frontal plane. Analysis of the behavior of the nonlinear coefficients in relation to the different planes reveals higher values for the frontal plane. This may be due to lateralization effects in individuals with osteoporotic spinal fractures, i.e., the trunk mobility of these patients was more reduced. Similar tendencies are shown by patients with chronic whiplash injury [52]. Raffalt et al. [53] showed that during eyes-closed trials, SampEn increased for young and elderly subjects in the AP direction, while SampEn increased in the ML direction for older individuals alone. The authors conclude that aging is associated with direction- and task-dependent changes in the dynamics of CoP movements executed during postural stance tasks. Our findings confirm that such properties are also present in individuals with osteoporotic spinal fractures.

In both groups, the noted increase in entropy during tasks with a higher degree of mechanical difficulty caused increased complexity, which means it is highly likely that additional control factors of the nervous system were activated. This can be interpreted as increased self-organization [14], and in this case serves as an improved effective strategy in postural control (no falls). Additionally, the observed increase in signal complexity may reflect an enhanced dynamic control of standing providing enhanced adaptability and flexibility in maintaining an erect posture with eyes closed; this explanation was proposed by Madeleine, Nielsen, and Arendt-Nielsen [52] for patients with chronic whiplash injury, who exhibited similar tendencies. A similar explanation was proposed by Sempere-Rubio et al. [54], where the study aim was to detect whether women with fibromyalgia syndrome (FMS) have altered postural control, and thus examining the sensory contribution to postural control. Thus, in both groups, an increase in entropy during more difficult balance tasks is possible because motor patterns adapt to this condition. However, it is worth emphasizing that individuals recovering from osteoporotic vertebral fractures were not able to complete all tasks, which suggests that their body cannot adapt to all conditions. Movement and stiffness are modified for fear of falling. This approach is also confirmed by increasing values of the fractal dimension. Blaszczyk and Klonowski [55] have shown that FD values increased with increasing task difficulty in healthy participants. Similar results were noted in studies based mainly on the traumatic injuries [52,56].

Overall, the results of our nonlinear measurements only partially confirmed Hypothesis 1. We have shown that individuals recovering from osteoporotic vertebral fractures are characterized by decreased irregularity, mainly in the medio-lateral direction, and reduced bodily complexity. Previous research published on the subject does not provide extensive information on postural control in individuals with osteoporotic spinal fractures; the majority of the extant research [38,57,58] shows that patients with osteoporosis, especially women, have greater anterior–posterior displacement of CoP. Our findings, in turn, indicate that individuals with osteoporotic spinal fractures tend to have a reduced ability to maintain balance under the conditions of reduced support surface area and with eliminated visual control. However, they manage well under dynamic conditions such as the fall risk test on a tilting Biodex plate. Therefore, we cannot accept the second hypothesis.

This study has some limitations. The group of individuals who experienced osteoporotic fractures of the spine is too small, primarily because not all individuals were able to pass the entire test protocol. Additionally, the control group may appear too young. On the other hand, such a young group guaranteed that the entire research protocol would be passed and that there would be no comorbidities associated with aging, for example. There are also the limitations of conducting multiple statistical tests on a relatively small and diverse sample size. There was no a priori power analysis, this being the first study to test patients who experienced osteoporotic fractures of the spine.

## Figures and Tables

**Figure 1 entropy-23-00375-f001:**
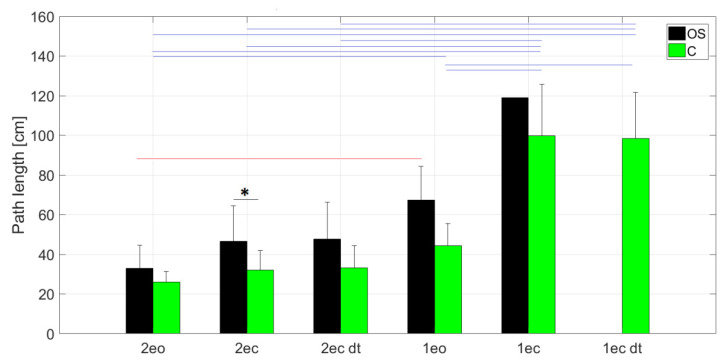
Path length values in subsequent trials for people with osteoporosis (OS) and the healthy group (C), where: * marks statistically significant differences between groups, the red line marks statistically significant differences between trials for OS group, and the blue line marks statistically significant differences between trials for the C group (*p* < 0.05).

**Figure 2 entropy-23-00375-f002:**
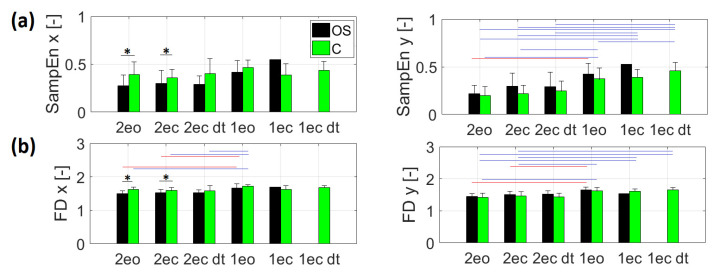
Coefficient values in subsequent trials for people with osteoporosis (OS) and healthy group (C): (**a**) sample entropy (SampEn); (**b**) fractal dimension (FD), where x—medio-lateral direction, y—anterior–posterior direction. * marks statistically significant differences between groups, the red line marks statistically significant differences between trials for the OS group, and the blue line marks statistically significant differences between trials for the C group (*p* < 0.05).

**Table 1 entropy-23-00375-t001:** Number of individuals who completed the static balance tasks in OS and C groups.

Task	Symbol	Number of Subjects Who Completed the OS/C Tasks
Standing on both legs with eyes open	2eo	17/17
Standing on both legs with eyes closed	2ec	16/17
Standing on both legs with eyes closed and with dual-task	2ec_dt	15/17
Standing on one leg with eyes open	1eo	8/17
Standing on one leg with eyes closed	1ec	1/17
Standing on one leg with eyes closed and with dual-task	1ec_dt	0/17

**Table 2 entropy-23-00375-t002:** Number of individuals from the OS group who completed the (FRT—Fall Risk Test) at a specific level, and data showing which subjects completed the static tasks, where an X means that the task was completed, and a dash (-) means that the task was not completed.

Number of Subjects Who Completed the FRT	Norm Level for the FRT Test	2eo	2ec	2ec_dt	1eo	1ec	1ec_dt
3	Fall	X	X	X	-	-	-
1	Fall	X	X	-	-	-	-
1	Fall	X	-	-	-	-	-
2	Fall	X	X	X	X	-	-
3	Norm	X	X	X	-	-	-
3	Norm	X	X	X	X	-	-
1	Below	X	X	X	X	X	-
2	Below	X	X	X	X	-	-
1	Below	X	X	X	-	-	-

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
