# Peer review of "The Impact of Visual Input and Support Area Manipulation on Postural Control in Subjects after Osteoporotic Vertebral Fracture"

_entropy, 2021, doi:10.3390/e23030375_

Round 1

Reviewer 1 Report

The paper at hand reports on a study about the effects of support area manipulation and visual changes to stability of standing on subjects who are recovering from a osteoporotic vertebral fracture.

The overall impression on the paper is positive. The study design is adequate. I appreciate the exploration of non-standard methods, in this case the non-linear measures to assess the center of pressure movement. However, in this work the employed measures are not well introduced, such that the paper is not self-contained. In addition, it feels as if the authors make use of these non-linear techniques without having understood what happens to the original data du to the non-linear transformation.

My recommendation to improve the paper: Add a (sub)section to materials and methods for each of the non-linear methods that are used, and explain how they are computed, what they compute and what they change to the data. In the Introduction a clear motivation is missing why non-linear tools are used, in this work. I do not think “we get significant numbers when using these methods” is a convincing argument – some support in reflecting about the modification of the original data is needed.

That said, I think this paper needs a major revision to address the points above.

Author Response

Thank you very much for your valuable comments. The introduction and the material and methods have been revised as suggested.

Reviewer 2 Report

The manuscript describes a study comparing several linear and non-linear variables characterizing postural control between osteoporotic vertebral fracture patients and healthy controls.

Overall, the manuscript is well structured, well written and carefully prepared. One limitation is, however, that the same information is often presented in the text and in figures or tables. Such repetitions should be avoided.

Unfortunately, there are several grave problems with the design, the methodology and the outcome interpretation of the underlying study. Therefore, I cannot recommend the paper for publication.

Main concerns

Study design: No hypotheses are stated. The study design is purely observational, it remains largely unclear, what research question the study should provide evidence for.

Methodologic 1: The sample size is small and is further compromised since not all participants were able to complete the study tasks. This hampers and compromises the statistical analysis.

Methodologic 2: The trial duration was very short (20 and 10 seconds). Reliable calculation of non-linear measures such as entropy or Lyapunov Exponent requires longer trials, typically 30 seconds.

Methodologic 3: During 10 second trials it is also not clear, if the prerequisite of stationarity (prerequisite for entropy or Lyapunov Exponent calculations) is met.  Often a long-duration drift can be observed in COP trajectories which would mean that this requirement was not met. Some studies differentiate the time series to remove underlying drifts, however, such a procedure was not mentioned in the paper.

Methodologic 4: A large number of statistical tests were calculated, however, no discussion of the associated alpha-error accumulation is found in the paper. A correction for this error would probably lead to mainly non-significant results.

Methodologic 5: p-values and effect sizes should be stated to make the study outcomes comparable with other data and to enable future meta-analyses.

Interpretation 1: There are statements in the discussion that are not in line with the data presented. For example, the concluding statement starts with the sentence “Overall, the results of our nonlinear measurements indicate that individuals recovering from osteoporotic vertebral fractures are characterized by increased irregularity and reduced bodily complexity”. – However, Figure 2 shows reduced complexity (SampEN) only for the x-direction. For the y direction consistently higher complexity is observed in the OS group (though not significant). Therefore, in my opinion the concluding statement is not sufficiently precise in its representation of the actual study outcomes.  

Author Response

Responses to Reviewer 2

The manuscript describes a study comparing several linear and non-linear variables characterizing postural control between osteoporotic vertebral fracture patients and healthy controls. Overall, the manuscript is well structured, well written and carefully prepared. One limitation is, however, that the same information is often presented in the text and in figures or tables. Such repetitions should be avoided. Unfortunately, there are several grave problems with the design, the methodology and the outcome interpretation of the underlying study. Therefore, I cannot recommend the paper for publication.

Main concerns:

Study design: No hypotheses are stated. The study design is purely observational, it remains largely unclear, what research question the study should provide evidence for.

Thank you very much for all comments. The purpose of the work was presented in the last sentence of the introduction. The hypotheses were not presented because the Editorial policy of the Journal does not require it. However, we added the hypotheses to the introduction section. Moreover, the introduction section has been enriched with new information.

Methodologic 1: The sample size is small and is further compromised since not all participants were able to complete the study tasks. This hampers and compromises the statistical analysis.

We agree with the Reviewer's comment. The sample size is small and this was also emphasized in the limitation of study (last paragraph of the discussion). However, we also would like to explain what caused such a small sample size. All participants qualified to the study group were operated by one surgeon and the number of them was limited due to organizational and financing procedures. Increasing the number of the study group would result in better statistical results, while the problem of the inability to perform tasks would not disappear. The surgeon, who operated those patients, expected such difficulties. We decided that homogeneity criterion (all participants operated by the same surgeon) is for us more important than improving the reliability of the statistical test. Thus, you can find a lot of papers where small number of persons is taken into account.

Methodologic 2: The trial duration was very short (20 and 10 seconds). Reliable calculation of non-linear measures such as entropy or Lyapunov Exponent requires longer trials, typically 30 seconds.

Thank you very much for this remark. We realize that trials are short and more reliable results would be obtained with trials of at least 30 seconds. However, in the literature you can find calculations for shorter time-series (20 seconds), e.g.

Szafraniec, R., J. Baranska, and M. Kuczynski. "Acute Effects of Core Stability Exercises on Balance Control." Acta Bioeng Biomech 20, no. 4 (2018): 145-51.

Stins, J. F., A. Ledebt, C. Emck, E. H. van Dokkum, and P. J. Beek. "Patterns of Postural Sway in High Anxious Children." Behav Brain Funct 5 (2009): 42.

Biec, E., J. Zima, D. Wojtowicz, B. Wojciechowska-Maszkowska, K. Krecisz, and M. Kuczynski. "Postural Stability in Young Adults with Down Syndrome in Challenging Conditions." PLoS One 9, no. 4 (2014): e94247.

Reinert, S. S., Jackson K., Bigelow K.,. "Using Posturography to Examine the Immediate Effects of Vestibular Therapy for Children with Autism Dpectrum Disorders: A Feasibility Study." Physical & Occupational Therapy in Pediatrics (2014): 1-16.

Schmit, J., M. Riley, S. Cummins-Sebree, L. Schmitt, and K. Shockley. "Children with Cerebral Palsy Effectively Modulate Postural Control to Perform a Supra-Postural Task." Gait Posture 42, no. 1 (2015): 49-53.

Schmit, J. M., M. Riley, S. Cummins-Sebree, L. Schmitt, and K. Shockley. "Functional Task Constraints Foster Enhanced Postural Control in Children with Cerebral Palsy." Phys Ther 96, no. 3 (2016): 348-54.

Terada, M., K. Kosik, N. Johnson, and P. Gribble. "Altered Postural Control Variability in Older-Aged Individuals with a History of Lateral Ankle Sprain." Gait Posture 60 (2018): 88-92.

Terada, M., M. Beard, S. Carey, K. Pfile, B. Pietrosimone, E. Rullestad, H. Whitaker, and P. Gribble. "Nonlinear Dynamic Measures for Evaluating Postural Control in Individuals with and without Chronic Ankle Instability." Motor Control 23, no. 2 (2019): 243-61.

Terada, M., K. D. Morgan, and P. A. Gribble. "Altered Movement Strategy of Chronic Ankle Instability Individuals with Postural Instability Classified Based on Nyquist and Bode Analyses." Clin Biomech (Bristol, Avon) 69 (2019): 39-43.

Krecisz, K., and M. Kuczynski. "Attentional Demands Associated with Augmented Visual Feedback During Quiet Standing." PeerJ 6 (2018): e5101.

In addition, we would like to emphasize that initially the protocols were set to 30 seconds. However, it turned out that the patients were unable to pass the trials. Thus, their duration was shortened so that research could be carried out. We did not want to give up the research because it seems interesting and it seems that nonlinear measures can help in the assessment of hidden structures. We understand that from the calculation point of view, our manuscript could be not fully valuable. On the other hand, we hope that the doctors who work with such patients will know that the impaired stability is something of a side effect.

Methodologic 3: During 10 second trials it is also not clear, if the prerequisite of stationarity (prerequisite for entropy or Lyapunov Exponent calculations) is met. Often a long-duration drift can be observed in COP trajectories which would mean that this requirement was not met. Some studies differentiate the time series to remove underlying drifts, however, such a procedure was not mentioned in the paper.

Thank you very much for this remark. According to: Yentes, J. M., N. Hunt, K. K. Schmid, J. P. Kaipust, D. McGrath, and N. Stergiou. "The Appropriate Use of Approximate Entropy and Sample Entropy with Short Data Sets." Ann Biomed Eng 41, no. 2 (2013): 349-65. Sample entropy is sensitive for very short data sets (N ≤ 200). I would like to emphasize that the data on which the calculations were made were much longer: 2000 point (data for trials of 20seconds) and 1000 points (data for trials of 10 seconds). Moreover Richman, J. S., and J. R. Moorman. "Physiological Time-Series Analysis Using Approximate Entropy and Sample Entropy." Am J Physiol Heart Circ Physiol 278, no. 6 (2000): H2039-49. suggested that SampEn is independent of data length and demonstrates relative consistency. However, Richman and Lake did state that when using data sets of less than 100, SampEn diverged from their predictions. In our paper the data were much longer.

When counting the fractal dimension, the algorithm was carefully selected. Some algorithms for dimensional complexity estimation, such as Katz’s one or D2, require a large number of samples to obtain reliable values. For this reason, it is necessary that the signal remains stationary during long duration intervals. This assumption is very difficult to achieve for physiological data. Nevertheless, Higuchi’s algorithm can be applied to short time series: Higuchi, T. "Approach to an Irregular Time Series on the Basis of the Fractal Theory." Physica D: Nonlinear Phenomena 31, no. 2 (1988): 277-83.

Lubetzky, Anat V., Daphna Harel, and Eyal Lubetzky. "On the Effects of Signal Processing on Sample Entropy for Postural Control." PLoS One 13, no. 3 (2018): e0193460-e60. wrote that differencing had a strong impact of the data. The same result was in our case. The values of SampEn were about 2-3 times as high as those produced by the position data. Nevertheless, non-stationarity is likely inherent in a biological process and a true physiological phenomenon. Therefore, Stergiou, N. Nonlinear Analysis for Human Movement Variability.: CRC Press, 2016 suggested that drift or trend should be removed only if there is a strong reason to believe that this drift is not part of the human performance pattern, that is if a calibration error or a device drift have occurred during data collection. We in our study did not have this problems.

After your remark, stationarity was checked in MatLab using the program using toolbox: Hristo Zhivomirov (2021). Stationarity Estimation of a Signal with Matlab (https://www.mathworks.com/matlabcentral/fileexchange/75118-stationarity-estimation-of-a-signal-with-matlab), MATLAB Central File Exchange. Retrieved February 16, 2021. We recalculated Lyapunov exponent for few persons for which the assumption was not met. It was resulted that there are no statistical differences between groups and trials. Therefore, it seems to us that it would be reasonable to remove the Lyapunov exponent from this work. Once again, I would like to thank you for this valuable remark.

Methodologic 4: A large number of statistical tests were calculated, however, no discussion of the associated alpha-error accumulation is found in the paper. A correction for this error would probably lead to mainly non-significant results.

Thank you for your remark. The large number of tests is due to the lack of normal distributions and different numbers of people in the OS group. In order to verify the correctness of the calculations, an ANOVA analysis with Bonferroni correction was used. So in this way we eliminated the associated alpha-error. I would like to emphasize that the results have not changed. However, the significance levels have changed.

Methodologic 5: p-values and effect sizes should be stated to make the study outcomes comparable with other data and to enable future meta-analyses.

Thank you, the values for the number p – values and effect size have been provided.

Interpretation 1: There are statements in the discussion that are not in line with the data presented. For example, the concluding statement starts with the sentence “Overall, the results of our nonlinear measurements indicate that individuals recovering from osteoporotic vertebral fractures are characterized by increased irregularity and reduced bodily complexity”. – However, Figure 2 shows reduced complexity (SampEN) only for the x-direction. For the y direction consistently higher complexity is observed in the OS group (though not significant). Therefore, in my opinion the concluding statement is not sufficiently precise in its representation of the actual study outcomes. 

Thank you very much, indeed the conclusion is too general not in line with the results. The interpretation has been improved.

Reviewer 3 Report

The Authors fail to furnish any information on the actual causes (origin) of those spinal fractures in their study group. Did those fractures result from a fall, or were they just the compression fractures totally unrelated to a fall? It may well be that those subjects had sustained the compression fractures and never experienced a fall, as is only too often the case with this type of fractures. That's why assessment of the falls-risk in them is not deemed essential, unless any such fractures were sustained through a fall.

No information is forthcoming either on how members of the control group had been recruited into the study. Were among them any individuals who had sustained falls resulting in any injuries? Were there any persons active in any kind of sports, etc. Was this a truly representative group for comparison purposes?

No specific inclusion criteria have been provided by the Authors.(Listing their basic characteristics might also help here).

The Authors do not provide any information about any foot dysfunctions in their study subjects, even though it is common knowledge any such dysfunctions are simply bound to affect the outcomes in this type of study.

A properly designed dual-task activity should entail appreciably more demanding  assignments in terms of cognitive complexity.

For instance, subtracting by 7 from a 100 would be a realistic cognitive load for effectively structured dual-task activity.

Multiplication table had been learnt by heart back at primary school, so it's hardly a cognitive challenge when one may readily fall back on one's memory.

There are no data on the cardiovascular status of the study subjects. Again, it's common knowledge that any cardiovascular complaints/conditions exert a major impact on the incidence of falls. Besides, they're quite common in the 65+ population, and appreciably less so in the 40+ group.

In my opinion the study population sample seems far too small for such far-reaching conclusions (please note there were also some drop-outs when the study protocol was already in progress).

As to the age difference between the study and the control groups, well, it's a generation gap, no less. Personally, I do think the Authors' argument in favour stretches the boundaries of academic credibility, without putting too fine a point on it...

ENGLISH LANGUAGE POINTS

The MS could well do with a through flushing-out by an English speaker, preferably well adept in editing academic papers, so that general flow of the discourse could be enhanced, common stock phrases orderly marshalled, and occasional grammatical deficiencies effectively purged from the text, otherwise pretty deftly put together in terms of its adequacy for the purpose at hand.

Please also note that 'falls-risk' (or falls risk) is now in prevalent use among the experts in the domain of falls prevention.

Here are two handy links for easy reference, just off the cuff:

https://www.physiopedia.com/Falls_Risk_Assessment_Tool_(FRAT):_An_Overview_to_Assist_Understanding_and_Conduction

https://www2.health.vic.gov.au/hospitals-and-health-services/patient-care/older-people/falls-mobility/falls/falls-risks

Author Response

Responses to Reviewer 3

The Authors fail to furnish any information on the actual causes (origin) of those spinal fractures in their study group. Did those fractures result from a fall, or were they just the compression fractures totally unrelated to a fall? It may well be that those subjects had sustained the compression fractures and never experienced a fall, as is only too often the case with this type of fractures. That's why assessment of the falls-risk in them is not deemed essential, unless any such fractures were sustained through a fall.

Thank you for this remark. All the individuals who took part in the research had compression fractures of the spine. These individuals reported that they had fallen several times in the past, but they were not serious falls. However, in the study group, a fall did not cause a spine fracture.

No information is forthcoming either on how members of the control group had been recruited into the study. Were among them any individuals who had sustained falls resulting in any injuries? Were there any persons active in any kind of sports, etc. Was this a truly representative group for comparison purposes? No specific inclusion criteria have been provided by the Authors. (Listing their basic characteristics might also help here).

The control group is a representative group. The inclusion criterion was no problems with balance.

The Authors do not provide any information about any foot dysfunctions in their study subjects, even though it is common knowledge any such dysfunctions are simply bound to affect the outcomes in this type of study.

All subjects, both those from the control group and those from the group of people after osteoporotic spine fracture, additionally underwent gait tests. Those who had correct gait parameters were taken for the balance analysis. However, since the people on the platform were tested barefoot, there was no noticeable foot dysfunction that could affect the test results. Patients also did not report such problems in their general history.

A properly designed dual-task activity should entail appreciably more demanding  assignments in terms of cognitive complexity. For instance, subtracting by 7 from a 100 would be a realistic cognitive load for effectively structured dual-task activity. Multiplication table had been learnt by heart back at primary school, so it's hardly a cognitive challenge when one may readily fall back on one's memory.

I agree that we learned the multiplication tables in school. We meant ‘multiplication task’ not ‘multiplication table’ and patients were asked to multiply two and three-digit numbers, which is not an easy task and many of them had trouble with that.

There are no data on the cardiovascular status of the study subjects. Again, it's common knowledge that any cardiovascular complaints/conditions exert a major impact on the incidence of falls. Besides, they're quite common in the 65+ population, and appreciably less so in the 40+ group.

The study participants had no cardiovascular problems. There was a surgeon and a medical doctor during patients’ examination and they were responsible for examining general health. Moreover, due to the methodology of the test (no change of position, rest between tests), these possible problems do not affect the test result, although, of course, problems with pressure in general increase the risk of a fall.

In my opinion the study population sample seems far too small for such far-reaching conclusions (please note there were also some drop-outs when the study protocol was already in progress). As to the age difference between the study and the control groups, well, it's a generation gap, no less. Personally, I do think the Authors' argument in favour stretches the boundaries of academic credibility, without putting too fine a point on it...

Thank you for this comment and we agree that the number of people surveyed is small for far-reaching conclusions. One of the inclusion criteria was that the operation was performed by the same surgeon to maintain homogeneity of the study group Additionally, in this study we wanted to point out that people with osteoporotic spine fractures have problems with balance. First of all, when the rehearsals start to get difficult, e.g. standing on one leg. (We had to reduce the duration of the trial). Therefore, increasing the number of people taking part in the research would certainly make the statistics more accurate, but then the problem “Inability to complete the task” that is highlighted in this paper would disappear. Moreover, the surgeon was not surprised by the fact, that these persons had problem with balance. Moreover it was resulted that they had low value of fall-risk test.

ENGLISH LANGUAGE POINTS

The MS could well do with a through flushing-out by an English speaker, preferably well adept in editing academic papers, so that general flow of the discourse could be enhanced, common stock phrases orderly marshalled, and occasional grammatical deficiencies effectively purged from the text, otherwise pretty deftly put together in terms of its adequacy for the purpose at hand.

Thank you very much for careful text checking. After the manuscript was modified, it was checked by a native speaker.

Please also note that 'falls-risk' (or falls risk) is now in prevalent use among the experts in the domain of falls prevention.

Here are two handy links for easy reference, just off the cuff:

https://www.physiopedia.com/Falls_Risk_Assessment_Tool_(FRAT):_An_Overview_to_Assist_Understanding_and_Conduction

https://www2.health.vic.gov.au/hospitals-and-health-services/patient-care/older-people/falls-mobility/falls/falls-risks

Thank you very much for valuable links. 

Round 2

Reviewer 1 Report

This version of the paper has clearly improved compared to the original submission. The authors much better motivate the use of the non-linear measures and introduce them adequate. Removing the Lyapunov exponent after the remarks of another reviewer and subsequent testing, helped to better focus the paper. Since the authors also addressed all other remarks by the reviewers, I think this paper is ready for publication.

Author Response

Thank you very much for your positive assessment and acceptance of the previous amendments and clarifications.

Reviewer 2 Report

The manuscript has improved on a number of points.

However, I’m still not satisfied with the statistical treatment. – If the prerequisites for an ANOVA are not met (normal distribution, etc.), then one cannot apply it. A Bonferroni correction could also be applied manually to the results of the non-parametric tests. If it was not justified to conduct ANOVAS before, then it is still not justified now.

Also, the study results are still not adequately represented in the discussion and conclusions, at least the last sentence of the abstract still does not adequately reflect all outcomes.  

Some arguments for a small sample size have been put forward, however, the study remains to be based on an inadequately small sample size. Particularly the consistency argument (one surgeon) is somewhat self-defeating, since this argument suggests that the study outcomes may not be generalizable to other surgeons, i.e. the reproducibility of the whole study is then questionable.  

Author Response

The manuscript has improved on a number of points.

However, I’m still not satisfied with the statistical treatment. – If the prerequisites for an ANOVA are not met (normal distribution, etc.), then one cannot apply it. A Bonferroni correction could also be applied manually to the results of the non-parametric tests. If it was not justified to conduct ANOVAS before, then it is still not justified now.

Thank you for this remark. We used non-parametric ANOVA Skillings-Mack test due to the fact that we had repeated measures in different conditions in each group. Additionally, we had big lost of data in OS group which led us to use Skillings-Mack test (not typical Friedman test) in this group. As a post-hoc test we chose Dunn test with Bonferroni correction. This statistic has asymptotically (for large sample sizes) normal distribution, and the p value is corrected by the number of possible simple comparisons in accordance with the selected correction. The legitimacy of using these tests is nicely explained in the articles below:

Chatfield, Mark, and Adrian Mander. "The Skillings-Mack Test (Friedman Test When There Are Missing Data)." The Stata journal 9, no. 2 (2009): 299-305.

Skillings, J.H., and G.A.  Mack. "On the Use of a Friedman-Type Statistic in Balanced and Unbalanced Block Designs." Technometrics 23 (1981): 171-77.

For inter-group comparison the non-parametric Mann-Whitney test was chosen.

Also, the study results are still not adequately represented in the discussion and conclusions, at least the last sentence of the abstract still does not adequately reflect all outcomes. 

Thank you for this remark. We corrected abstract and discussion, so that it reflect all outcomes.

Some arguments for a small sample size have been put forward, however, the study remains to be based on an inadequately small sample size. Particularly the consistency argument (one surgeon) is somewhat self-defeating, since this argument suggests that the study outcomes may not be generalizable to other surgeons, i.e. the reproducibility of the whole study is then questionable.

The reliability standards of the studies in medicine require to qualify to one study patients operated by one surgeon highly qualified and experienced in described type of surgery. Sometimes it is acceptable to qualified patients operated by two or more highly experienced surgeons but in this case it is necessary that all of them use the same techniques and procedures during surgery.

Patients operated by only one surgeon were qualified to take part in our study, because he was the only one person highly experienced in minimally invasive stabilization surgery of the spine. We would like to emphasize that the analysis of the surgery technique was not the aim of this manuscript.

Reviewer 3 Report

The Reviewer has no further objections, nor any extra comments to the revised paper.

Author Response

(The authors gave the same response as above.)
